# Data Mining and Validation of AMPK Pathway as a Novel Candidate Role Affecting Intramuscular Fat Content in Pigs

**DOI:** 10.3390/ani9040137

**Published:** 2019-04-01

**Authors:** Chaogang Yao, Daxin Pang, Chao Lu, Aishi Xu, Peixuan Huang, Hongsheng Ouyang, Hao Yu

**Affiliations:** Jilin Provincial Key Laboratory of Animal Embryo Engineering, College of Animal Sciences, Jilin University, Changchun 130062, China; yaocg13@mails.jlu.edu.cn (C.Y.); pdx@jlu.edu.cn (D.P.); swjs092408147@163.com (C.L.); xuaishi2008@aliyun.com (A.X.); huangpx16@mails.jlu.edu.cn (P.H.)

**Keywords:** AMPK signaling pathway, pig, intramuscular fat, data mining

## Abstract

**Simple Summary:**

Intramuscular fat (IMF) is increasingly being recognized as a key meat trait in the modern pork industry. The aims of this research were to identify potential signaling pathways associated with IMF content in the longissimus dorsi (LD) muscle of different pig breeds and investigate the gene expression levels in the screened signaling pathways. Our results indicated that the AMPK signaling pathway may be related to IMF deposition in the LD muscle of pigs. The results of qRT-PCR analysis showed that the expression of ten key hub genes (*AMPK, ADIPOR1, ADIPOR2, LKB1, CAMKKβ, CPT1A, CPT1B, PGC-1α, CD36,* and *ACC1*) differed between the LD muscle of Min and Large White pigs. The protein expression levels of AMPK, LKB1, CaMKK2, CPT1A, and ACC1 were similar to the genes expression patterns in the LD muscle of Large White pigs. The results of this study provide novel insights into the regulatory function of the AMPK signaling pathway in relation to the IMF content in the LD muscle of different pigs.

**Abstract:**

Intramuscular fat (IMF) is an important economic trait for pork quality and a complex quantitative trait regulated by multiple genes. The objective of this work was to investigate the novel transcriptional effects of a multigene pathway on IMF deposition in the longissimus dorsi (LD) muscles of pigs. Potential signaling pathways were screened by mining data from three gene expression profiles in the Gene Expression Omnibus (GEO) database. We designed quantitative real-time reverse transcription-polymerase chain reaction (qRT-PCR) arrays for the candidate signaling pathways to verify the results in the LD muscles of two pig breeds with different IMF contents (Large White and Min). Western blot analysis was used to detect the expression levels of several candidate proteins. Our results showed that the AMPK signaling pathway was screened via bioinformatics analysis. Ten key hub genes of this signaling pathway (*AMPK, ADIPOR1, ADIPOR2, LKB1, CAMKKβ, CPT1A, CPT1B, PGC-1α, CD36,* and *ACC1*) were differentially expressed between the Large White and Min pigs. Western blot analysis further confirmed that LKB1/CaMKK2-AMPK-ACC1-CPT1A axis dominates the activity of AMPK signaling pathway. Statistical analyses revealed that AMPK signaling pathway activity clearly varied among the two pig breeds. Based on these results, we concluded that the activation of the AMPK signaling pathway plays a positive role in reducing IMF deposition in pigs.

## 1. Introduction

As one of the most important domesticated animals for agricultural production, pigs provide many meat products for humans [1]. In modern society, pork quality has had an increasing influence on consumer acceptance and initial purchasing decisions. Consumers are interested in several major pork quality traits, including meat color, pH value, water holding capacity, and intramuscular fat (IMF) content, which are becoming increasingly important from an economic perspective [2,3]. Skeletal muscle is a heterogeneous tissue that consists of different types of myofibers, connective tissue, vascular tissue, nervous tissue, and IMF [4]. IMF is a major meat quality trait in pigs, and its content is directly associated with the sensory qualities, flavor, juiciness, tenderness, and nutritional quality of pork [5,6]. In recent decades, several studies have focused on the relationship between IMF and pork quality [7,8,9,10].

Famous lean pig breeds, such as Large White, Landrace, and Duroc, have lower IMF contents and reduced meat quality due to the intensive selection processes used to improve pork productivity. However, many excellent indigenous breeds are distributed in China, such as the Jinhua, Laiwu, Meishan, and Min, and they have higher IMF contents and better meat quality than the lean breeds [11,12,13,14,15,16]. Thus, it will be beneficial to reveal the molecular mechanisms of IMF deposition by comparing gene expression between lean and indigenous Chinese pig breeds.

With the rapid development of microarray and RNA-seq technologies in the last few decades, researchers are now able to study many differentially expressed genes (DEGs) simultaneously in a given tissue. To date, many studies concerning meat quality traits and gene expression in pigs have been reported [17,18,19]. Fortunately, the relevant datasets have been deposited and stored in the National Center for Biotechnology Information (NCBI) Gene Expression Omnibus (GEO) database and are freely accessible to researchers worldwide. However, few studies have focused on integrating and reanalyzing these datasets, which contain valuable clues regarding important porcine economic traits. Thus, by integrating and reanalyzing these datasets, we can provide significant insights into the molecular changes associated with IMF deposition.

In this study, we integrated and reanalyzed three original expression profiles from the GEO database based on a current popular differential gene expression analysis method. We found that the AMPK pathway plays a critical role in IMF deposition. We further validated this pathway through the use of quantitative real-time reverse transcription-polymerase chain reaction (qRT-PCR) arrays in the Large White and Min pig breeds.

## 2. Materials and Methods 

### 2.1. GEO Data Collection

The gene expression profiles *GSE24192*, *GSE75045*, and *GSE99092* [12,20,21] were downloaded from the GEO database. The *GSE24192* dataset contained six samples, which included three Large White longissimus dorsi (LD) samples and three Northeastern Indigenous (Min) LD samples. The *GSE75045* dataset contained six samples, which included three Large White LD samples and three Wannanhua LD samples. The *GSE99092* dataset contained six samples, which included three Large White LD samples and three Wei LD samples. In the present study, Large White was set as the experimental group, the indigenous Chinese pig breeds were set as the control group.

### 2.2. Identification of DEGs

The limma package from Bioconductor and the online tool iDEP (https://github.com/gexijin/iDEP) were used to identify DEGs for the selected gene expression profile datasets [22]. A *p*-value less than 0.05 and |log_Fold Change(FC)_| ≥1 were regarded as the cutoff thresholds for DEGs. 

### 2.3. Signaling Pathway Enrichment Analysis of DEGs

To analyze the functions of the DEGs, we performed a Kyoto Encyclopedia of Genes and Genomes (KEGG) pathway analysis of the DEGs using the online tool Kobas 3.0 [23]. A *p*-value less than 0.05 was considered statistically significant. ClusterProfiler was used for the statistical analysis and visualization of the functional profiles of the DEGs in the GEO datasets and qRT-PCR arrays [24]. TBtools was used to construct the Venn diagrams of the KEGG pathways for the DEGs in the three GEO datasets.

### 2.4. Animals and Tissue Collection

Three sows each from the Min and Large White breeds were used in this study. The Min pig is an excellent indigenous breed from northeastern China, and it has an IMF content higher than that of Large White pigs [12]. The pigs used in this study were obtained from the Institute of Animal Husbandry Research, Heilongjiang Academy of Agricultural Sciences (Harbin, China). The pigs were raised for 180 days under the same conditions. When the pigs were slaughtered, the LD muscle was collected between the 10th and 12th ribs from the carcasses. All tissue samples were divided into two groups; one group was quickly frozen immediately after collection and stored at −80 °C until use in qRT-PCR arrays, and the other group was stored at 4 °C for the determination of IMF content. All animal procedures were performed according to the University Committee on the Use and Care of Animals at Jilin University (approval ID: 201706030).

### 2.5. Determination of IMF Content 

In the present study, the IMF content was measured in each LD sample by the Soxhlet extraction method with petroleum ether [25].

### 2.6. RNA Extraction and Quantitative Real-Time Reverse Transcription-Polymerase Chain Reaction (Qrt-PCR) Arrays

Total RNA from the LD was isolated from approximately 200 mg of frozen tissue using TRIzol-A^+^ (TIANGEN, Beijing, China) following the manufacturer’s instructions. The BioRT cDNA First Strand Synthesis Kit (Bioer Technology, Hangzhou, China) was used to synthesize first-strand cDNA. Subsequently, the expression levels of the target genes were analyzed on an iQ™5 real-time PCR detection system (Bio-Rad, USA). A BioEasy SYBR Green I Real Time PCR kit (Bioer Technology) was used according to the manufacturer’s instructions to detect each sample in triplicate. The primers used for the qRT-PCR arrays are listed in Appendix A. The gene IDs from the selected pathways were obtained from the KEGG database (Appendix A).

### 2.7. Western Blot Analysis

Protein samples from the LD of Large White (*n* = 3) and Min (*n* = 3) were separated by 10% SDS-PAGE gels and transferred electrophoretically onto polyvinylidene fluoride (PVDF) membranes (Millipore, Boston, MA, USA). Western blot was performed with rabbit anti-AMPK alpha 1 polyclonal antibody (1:1000, Bioss, Beijing, China), rabbit anti-LKB1 polyclonal antibody (1:1000, Bioss), rabbit anti-CaMKK2 polyclonal antibody (1:1000, Bioss), rabbit anti-CPT1A polyclonal antibody (1:1000, Bioss), and rabbit anti-ACC1 polyclonal antibody (1:1000, WUHAN SANYING) as the primary antibodies and Horseradish peroxidase (HRP)-conjugated goat anti-rabbit antibody as the second antibody. Rabbit anti-β-actin polyclonal antibody (1:8000, Bioss) was used as an internal control. Signals were detected using the SuperSignal WestPico Chemiluminescent Substrate Kit (Thermo Fisher Scientific) according to the manufacturer’s instructions.

### 2.8. Statistical Analysis of the Qrt-PCR Array Results

In the present study, the Large White pigs were set as the experimental group and the Min pigs were set as the control group. GraphPad Prism 6.01 (GraphPad Software, San Diego, CA, USA) was used for analyzing our results. Student’s *t*-tests were used to compare the control and experimental groups. For all comparisons, * *p* < 0.05, ** *p* < 0.01, *** *p* < 0.001 and **** *p* < 0.0001 were considered significant differences. The 2^−ΔCT^ method was used to calculate the Ct values from the qRT-PCR array data. The limma package was used to identify the significant DEGs of the qRT-PCR arrays between different groups [22]. TBtools was used to draw heatmaps of the qRT-PCR arrays (https://github.com/CJ-Chen/TBtools). The online tool KEGG Mapper was used to draw the colored map of DEGs (https://www.kegg.jp/kegg/mapper.html). The STRING database was used to predict protein interactions and construct the network for DEGs [26]. The protein-protein interaction (PPI) network was visualized by Cytoscape [27].

## 3. Results

### 3.1. Identification of DEGs in GEO Datasets

According to the cutoff threshold (*p* < 0.05 and |log_FC_| ≥ 1), in *GSE24192*, 1237 DEGs were identified in the LD of Large White pigs when compared with the indigenous Chinese breeds, and they included 877 upregulated genes and 360 downregulated genes. In *GSE75045*, a total of 2582 DEGs were identified in the LD of Large White pigs, and they included 1096 upregulated genes and 1486 downregulated genes. Finally, in *GSE99092*, a total of 1822 DEGs were identified in the LD of Large White pigs, and they included 809 upregulated genes and 1013 downregulated genes. 

### 3.2. Pathway Enrichment of DEGs in GEO Datasets

The KEGG pathway enrichment results (Figure 1D,E; Table 1, Table 2 and Table 3) yielded no shared pathways among the downregulated DEGs of Large White pigs in the three GEO datasets. In contrast, the AMPK signaling pathway (ssc04152), the peroxisome proliferator-activated receptor (PPAR) signaling pathway (ssc03320), fat digestion and absorption (ssc04975), fatty acid metabolism (ssc01212), metabolic pathways (ssc01100), and biosynthesis of amino acids (ssc01230) were among the upregulated DEGs in Large White pigs in the three GEO datasets. The AMPK signaling pathway may represent a novel pathway for regulating IMF deposition in pigs.

### 3.3. IMF Content of LD Muscles in the Two Pigs

As shown in Figure 2, the Min pigs had significantly higher IMF content than the Large White pigs.

### 3.4. Validation of the AMPK Signaling Pathway in the LD Muscles of the Two Pig Breeds

The AMPK signaling pathway (ssc04152) consists of 117 genes (Appendix A). In this study, 114 of these genes were validated via qRT-PCR array. Of them, 40 genes were differentially expressed in the Large White LD; 22 were upregulated and 18 were downregulated. The qRT-PCR results for the AMPK signaling pathway are shown in Figure 3, Figure 4 and Figure 5, Table 4 and Table 5. A heatmap of the AMPK signaling pathway is presented in Figure 3A, indicating that the expression patterns of the DEGs in the AMPK signaling pathway among the Large White and Min breeds show significant differences. Figure 3B shows a colored map of the AMPK signaling pathway in the LD of Large White pigs. Figure 4 shows a PPI network of 14 upregulated genes in the LD muscle of Large White pigs, and it contains 14 nodes and 36 edges. The most significant 10 node degree genes are *SLC2A4, PGC-1α, LKB1, AMPK, LIPE, CD36, CaMKK2, FOXO3, FOXO1,* and *CPT1B* (Table 5). Moreover, based on the results of the colored map and PPI network, 10 genes in the AMPK signaling pathway (*AMPK, ADIPOR1, ADIPOR2, LKB1, CaMKKβ, CPT1A, CPT1B, PGC-1α, CD36,* and *ACC1*) were selected as key hub genes, and their expression patterns are presented in Figure 5. These hub genes are associated with fatty acid oxidation, lipid oxidation, and fatty acid metabolic processes. In addition, the protein expression levels of AMPK, LKB1, CaMKK2, CPT1A, and ACC1 were examined in the LD muscle of the Min and Large White pigs by Western blot analysis. The protein expression levels of AMPK, LKB1, CaMKK2, and CPT1A were higher in the Large White group than that the Min group, ACC1 showed low expression level in the LD muscle of Large White pigs (Figure 6). Taken together, these results show that the AMPK signaling pathway is more active in the Large White breed than the Min breeds.

### 3.5. GO Enrichment of DEGs in Qrt-PCR Arrays

The biological processes encoded by upregulated genes were involved in fatty acid oxidation, lipid oxidation, and fatty acid metabolic processes, while the biological processes encoded by the downregulated genes were involved in carbohydrate metabolic processes, including glucose, hexose, and monosaccharide metabolism, as well as hexose and monosaccharide catabolism (Figure 7). These results indicate that compared with the LD of Min pigs, the LD of Large White pigs consumes more fat for energy metabolism than carbohydrates.

## 4. Discussion

In the modern pork industry, the IMF content is an important trait that is positively associated with pork quality and in demand by consumers. As a complex meat trait, IMF deposition in the LD muscle is regulated by multiple genes and pathways. In this study, by integrating and reanalyzing three gene expression profiles, we compared the pathways related to IMF deposition in the LD muscle of Large White pigs with those of indigenous breeds. Several candidate signaling pathways were found, and the expression patterns of genes in the AMPK pathway in pigs were validated by qRT-PCR arrays in subsequent experiments.

The AMPK signaling pathway plays critical roles in controlling both glucose and lipid metabolism. *AMPK* is the central gene of the AMPK signaling pathway and a heterotrimeric enzyme with α, β, and γ subunits. Once activated, *AMPK* promotes lipid oxidation and glucose uptake, inhibits lipid synthesis and decreases IMF contents [28]. Accordingly, the activity of *AMPK* is inversely correlated with IMF accumulation. In the present study, *AMPK* was highly expressed in the LD muscle of Large White pigs, which has a lower IMF content than Min pigs. This result is consistent with several previous reports demonstrating that the expression levels of *AMPK* are higher in the low-IMF-content skeletal muscle of cattle and sheep [29,30,31], suggesting that *AMPK* plays a positive role in reducing the IMF content in pigs.

As a member of the adipocytokines, adiponectin plays crucial roles in whole-body energy homeostasis by stimulating *AMPK. ADIPOR1* and *ADIPOR2* are two major receptors for adiponectin and play key roles in metabolic pathways that regulate glucose and lipid metabolism, inflammation, and oxidative stress [32]. *ADIPOR1* and *ADIPOR2* mediate the metabolic actions of adiponectin by activating *AMPK* and *PPARα*, respectively. This activation leads to increased fatty acid oxidation and glucose uptake in mice [33,34,35]. Moreover, muscle-specific disruption of *ADIPOR1* inhibits the increase in intracellular Ca^2+^ concentration and reduces the activation of calmodulin-dependent kinase β (*CaMKKβ*) and *AMPK* by adiponectin. Consistent with these previously reported results, *ADIPOR1* and *ADIPOR2* were both upregulated in the LD muscle of Large White pigs in the present study, suggesting that these two genes may account for the low IMF accumulation in pigs. Interestingly, two upstream kinases of *AMPK*, the tumor suppressor *LKB1* and Ca^2+^/*CaMKKβ*, which participate in the phosphorylation and activation of *AMPK*, were simultaneously highly expressed in the LD muscle of Large White pigs. *AMPK* can be activated via two distinct mechanisms, *LKB1* encodes a serine-threonine kinase that directly phosphorylates and activates AMPK, and CaMKKβ can form a complex with and activate AMPK through their kinase domains in skeletal muscle [36,37,38,39]. Consistent with these previous findings, our results suggest that *LKB1* and *CaMKKβ* play critical roles in reducing the IMF content in pigs by activating the expression of *AMPK*.

Furthermore, *AMPK* plays a central role in controlling lipid metabolism by regulating the downstream acetyl-CoA carboxylase (ACC1) and carnitine palmitoyltransferase 1 (CPT1) pathways. *CPT1*, a rate-limiting enzyme of mitochondrial fatty acid β-oxidation, is closely associated with fat deposition. Additionally, *CPT1A* and *CPT1B*, two common *CPT1* subtypes in mammals, play prominent roles in fatty acid oxidation and lipid accumulation in humans, chickens, and pigs [40,41,42]. Moreover, according to the KEGG database, *CPT1A* and *CPT1B* are involved in the AMPK signaling pathway, implying that these genes participate in the mediation of fatty acid oxidation. In addition, the *ACC1* gene encodes acetyl-CoA carboxylase (*ACC*), which is the rate-limiting enzyme responsible for the de novo synthesis of fatty acids [43]. As a target gene of the AMPK signaling pathway, the activity of *ACC1* is inhibited by *AMPK*. Similarly, an increase in activity of *ACC1* can also inhibit the expression of *CPT1* and fatty acid oxidation in skeletal muscles [44,45,46]. Consistent with these previous results, both *CPT1A* and *CPT1B* were highly expressed and *ACC1* was significantly downregulated in the LD muscle of Large White pigs, suggesting that the AMPK-ACC1-CPT1 pathway was positively associated with a decrease in fatty acid synthesis and IMF deposition in pigs. 

In addition to *CPT1*, the fatty acid transporter fatty acid translocase/cluster of differentiation 36 (*FAT/CD36*) has also been found to regulate FA oxidation in skeletal muscle of human and mice [47,48]. *FAT/CD36* has been identified as contributing to fatty acid transport and oxidation in mice [49,50,51]. In our results, *FAT/CD36* was highly expressed in the LD muscle of Large White pigs, implying that the expression of this gene has a negative effect on IMF deposition in Large White pigs. Interestingly, the peroxisome proliferator-activated receptor gamma coactivator 1-alpha (*PPARGC1A, PGC-1α*) was also involved in the activation of the *CPT1* gene [52]. *PGC-1α* plays an important role in glucose and fatty acid metabolism and has a negative relationship with the IMF content in pigs [53,54,55]. In this study, *PGC-1α* presented a high expression level in the LD of Large White, suggesting that this gene exerts a negative effect on IMF deposition in pigs. Additionally, the protein expression of AMPK, LKB1, CaMKK2, CPT1A, and ACC1 were further analyzed with Western blot. In our results, four proteins (AMPK, LKB1, CaMKK2, and CPT1A) showed higher expression in the LD muscle of Large White pigs than that in Min pigs, and ACC1 had a low protein expression level in the Large White group, indicating that the AMPK signaling pathway is more active at the protein level and that the oxidative degradation of fatty acids is stronger in the Large White group. In summary, the above results suggest that the activation of the AMPK signaling pathway reduces the IMF content in the LD muscle of Large White pigs.

## 5. Conclusions

In conclusion, this study illustrates that the accumulation of IMF in Large White pigs is related to activation of the AMPK signaling pathway. The relatively high expression of genes in the AMPK pathway may represent one of the more significant features of pigs with artificially lean meat. Our results are also helpful for interpreting the different molecular mechanisms of IMF deposition between lean and fat pig breeds.

## Figures and Tables

**Figure 1 animals-09-00137-f001:**
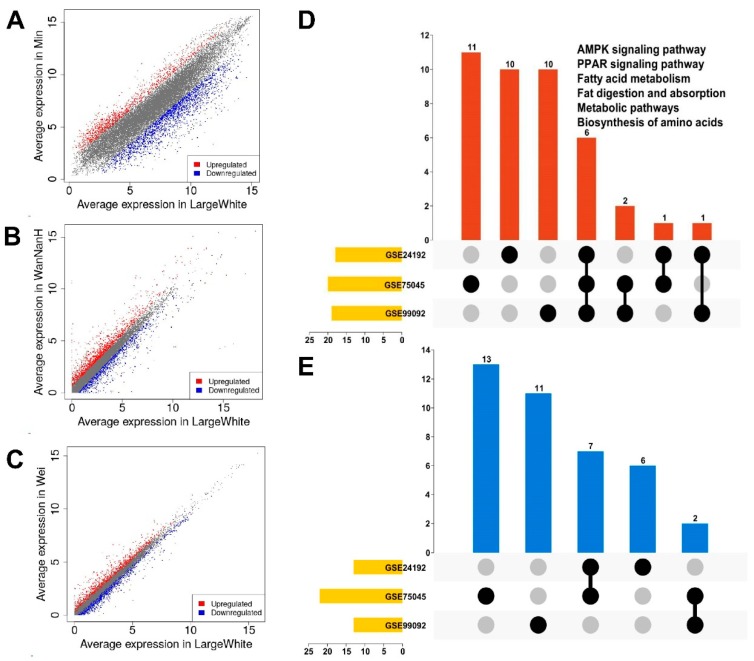
Data mining results for the three Gene Expression Omnibus (GEO) datasets. Scatter plot of differentially expressed genes (DEGs) in (**A**) *GES24192,* (**B**) *GSE75045*, and (**C**) *GSE99092*. (**D**) Venn diagrams of the Kyoto Encyclopedia of Genes and Genomes (KEGG) pathways for the upregulated genes in the three GEO datasets. The six shared pathways (AMPK signaling pathway, the peroxisome proliferator-activated receptor (PPAR) signaling pathway, fat digestion and absorption, fatty acid metabolism, metabolic pathways, and biosynthesis of amino acids) are listed in the figure. (**E**) Venn diagrams of the KEGG pathways for the downregulated genes in the three GEO datasets.

**Figure 2 animals-09-00137-f002:**
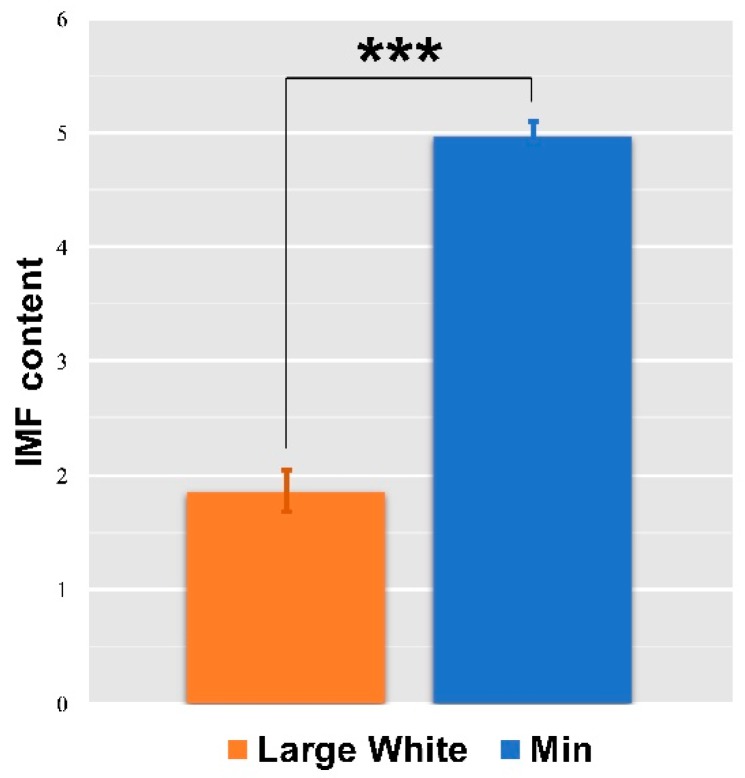
Detection of the IMF content in the LD muscles of Large White and Min pigs. Data represent the means ± SEM (*n* = 3), *** *p* < 0.001.

**Figure 3 animals-09-00137-f003:**
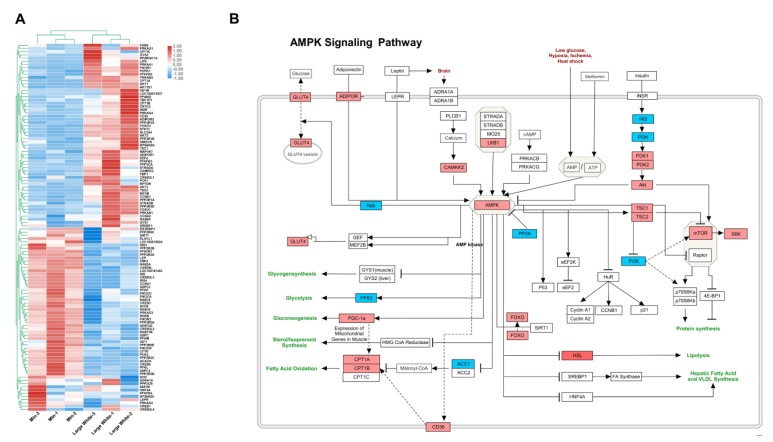
qRT-PCR array results in two pig breeds. (**A**) Heatmap of all qRT-PCR array genes. (**B**) Colored map of the AMPK signaling pathway. Upregulated and downregulated genes are indicated by red and blue, respectively.

**Figure 4 animals-09-00137-f004:**
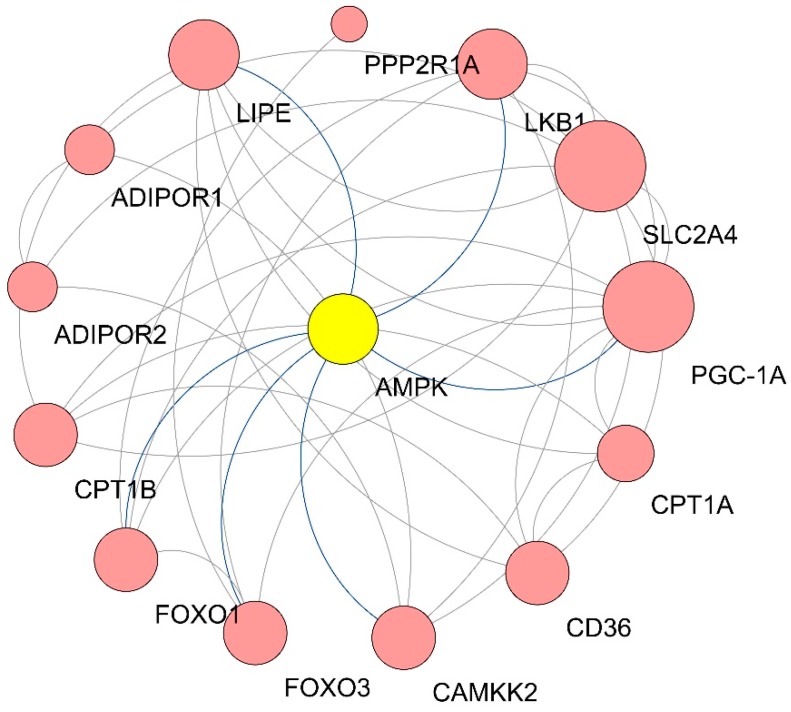
PPI network for the upregulated genes in the AMPK signaling pathway. The nodes represent the proteins (genes); the edges represent the interaction of proteins (genes).

**Figure 5 animals-09-00137-f005:**
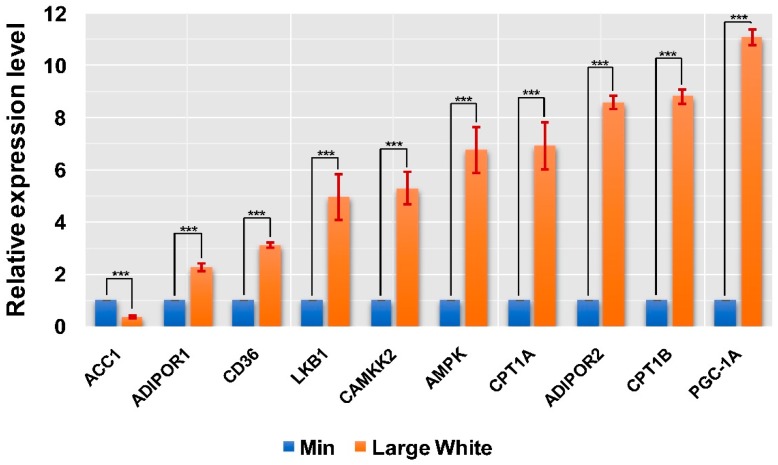
Relative expression levels of ten key hub genes in the AMPK signaling pathway in two pig breeds. All data are shown as the means ± SEM (*n* = 3), *** *p* < 0.001.

**Figure 6 animals-09-00137-f006:**
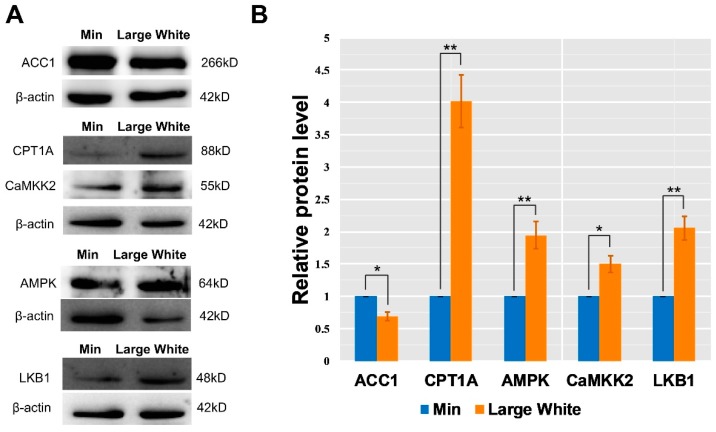
Relative expression levels of AMPK, LKB1, CaMKK2, CPT1A, and ACC1 proteins in the AMPK signaling pathway in two pig breeds. All data are shown as the means ± SEM (*n* = 3), * *p* < 0.05, ** *p* < 0.01. Representative western blots (**A**) and quantitative densitometry analysis (**B**) of AMPK, LKB1, CaMKK2, CPT1A, and ACC1 protein levels are shown in the LD muscle of Large White and Min pigs.

**Figure 7 animals-09-00137-f007:**
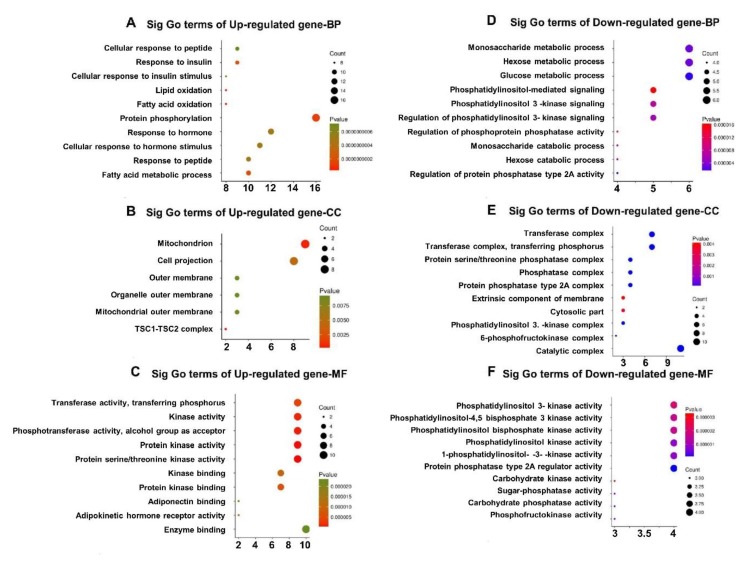
GO terms include three complementary biological roles, Biological Process (BP), Molecular Function (MF), and Cellular Component (CC), for the DEGs in the qRT-PCR array. BP for upregulated (**A**) and downregulated (**D**) genes. CC for upregulated (**B**) and downregulated (**E**) genes. MF for upregulated (**C**) and downregulated (**F**) genes.

**Table 1 animals-09-00137-t001:** KEGG pathway enrichment of the DEGs in the Large White pigs from *GSE24912*.

Pathway ID	Name	Gene Count	*p*-Value
**Upregulated DEGs**			
ssc04145	Phagosome	36	1.66 × 10^−22^
ssc01100	Metabolic pathways	87	2.06 × 10^−18^
ssc00100	Steroid biosynthesis	6	3.08 × 10^−5^
ssc00330	Arginine and proline metabolism	8	4.48 × 10^−5^
ssc04062	Chemokine signaling pathway	15	5.27 × 10^−5^
ssc00590	Arachidonic acid metabolism	9	5.59 × 10^−5^
ssc01230	Biosynthesis of amino acids	9	8.49 × 10^−5^
ssc04923	Regulation of lipolysis in adipocytes	8	1.32 × 10^−4^
ssc04060	Cytokine-cytokine receptor interaction	16	1.92 × 10^−4^
ssc00010	Glycolysis/gluconeogenesis	8	2.24 × 10^−4^
ssc04810	Regulation of actin cytoskeleton	15	2.75 × 10^−4^
ssc04390	Hippo signaling pathway	12	2.98 × 10^−4^
ssc00592	alpha-Linolenic acid metabolism	5	4.30 × 10^−4^
ssc03320	PPAR signaling pathway	8	4.72 × 10^−4^
ssc04152	AMPK signaling pathway	10	9.76 × 10^−4^
ssc04975	Fat digestion and absorption	5	2.28 × 10^−3^
ssc00071	Fatty acid degradation	5	4.66 × 10^−3^
ssc01212	Fatty acid metabolism	5	7.78 × 10^−3^
**Downregulated DEGs**			
ssc00190	Oxidative phosphorylation	17	1.67 × 10^−13^
ssc01100	Metabolic pathways	38	5.02 × 10^−10^
ssc01210	2-Oxocarboxylic acid metabolism	4	5.24 × 10^−5^
ssc04024	cAMP signaling pathway	10	5.68 × 10^−5^
ssc01230	Biosynthesis of amino acids	5	7.29 × 10^−4^
ssc04960	Aldosterone-regulated sodium reabsorption	4	9.76 × 10^−4^
ssc04931	Insulin resistance	6	1.03 × 10^−3^
ssc00220	Arginine biosynthesis	3	1.33 × 10^−3^
ssc04923	Regulation of lipolysis in adipocytes	4	2.79 × 10^−3^
ssc00400	Phenylalanine, tyrosine, and tryptophan biosynthesis	2	3.14 × 10^−3^
ssc03320	PPAR signaling pathway	4	5.59 × 10^−3^
ssc04920	Adipocytokine signaling pathway	4	5.86 × 10^−3^
ssc00250	Alanine, aspartate, and glutamate metabolism	3	6.59 × 10^−3^

**Table 2 animals-09-00137-t002:** KEGG pathway enrichment of the DEGs in the Large White pigs from *GSE75045*.

Pathway ID	Name	Gene Count	*p*-Value
**Upregulated DEGs**			
ssc04922	Glucagon signaling pathway	18	2.68 × 10^−7^
ssc04910	Insulin signaling pathway	20	5.21 × 10^−6^
ssc00010	Glycolysis/gluconeogenesis	13	9.46 × 10^−6^
ssc01230	Biosynthesis of amino acids	14	1.18 × 10^−5^
ssc01200	Carbon metabolism	14	5.35 × 10^−4^
ssc04152	AMPK signaling pathway	14	1.01 × 10^−3^
ssc00500	Starch and sucrose metabolism	8	2.01 × 10^−3^
ssc00760	Nicotinate and nicotinamide metabolism	6	2.10 × 10^−3^
ssc04974	Protein digestion and absorption	10	2.29 × 10^−3^
ssc04911	Insulin secretion	9	4.71 × 10^−3^
ssc04931	Insulin resistance	12	4.82 × 10^−3^
ssc00051	Fructose and mannose metabolism	6	6.50 × 10^−3^
ssc01100	Metabolic pathways	67	8.00 × 10^−3^
ssc00620	Pyruvate metabolism	6	8.50 × 10^−3^
ssc04022	cGMP-PKG signaling pathway	14	9.21 × 10^−3^
ssc04015	Rap1 signaling pathway	16	9.32 × 10^−3^
ssc00030	Pentose phosphate pathway	5	9.47 × 10^−3^
ssc01212	Fatty acid metabolism	7	1.37 × 10^−2^
ssc03320	PPAR signaling pathway	6	4.81 × 10^−2^
ssc04975	Fat digestion and absorption	4	4.83 × 10^−2^
**Downregulated DEGs**			
ssc01100	Metabolic pathways	127	1.26 × 10^−9^
ssc00071	Fatty acid degradation	16	1.62 × 10^−7^
ssc01212	Fatty acid metabolism	16	2.91 × 10^−7^
ssc03320	PPAR signaling pathway	16	2.16 × 10^−5^
ssc00190	Oxidative phosphorylation	23	2.84 × 10^−5^
ssc04146	Peroxisome	18	5.44 × 10^−5^
ssc04260	Cardiac muscle contraction	15	1.18 × 10^−4^
ssc00640	Propanoate metabolism	9	1.86 × 10^−4^
ssc00620	Pyruvate metabolism	10	3.69 × 10^−4^
ssc00280	Valine, leucine, and isoleucine degradation	11	5.68 × 10^−4^
ssc01200	Carbon metabolism	17	1.05 × 10^−3^
ssc00480	Glutathione metabolism	10	1.14 × 10^−3^
ssc00561	Glycerolipid metabolism	10	2.15 × 10^−3^
ssc00270	Cysteine and methionine metabolism	9	4.87 × 10^−3^
ssc00250	Alanine, aspartate, and glutamate metabolism	8	5.33 × 10^−3^
ssc00062	Fatty acid elongation	5	7.17 × 10^−3^
ssc04261	Adrenergic signaling in cardiomyocytes	16	1.20 × 10^−2^
ssc04810	Regulation of actin cytoskeleton	21	1.53 × 10^−2^
ssc00061	Fatty acid biosynthesis	4	1.53 × 10^−2^
ssc04920	Adipocytokine signaling pathway	10	1.82 × 10^−2^
ssc00400	Phenylalanine, tyrosine and tryptophan biosynthesis	3	1.88 × 10^−2^
ssc01230	Biosynthesis of amino acids	9	4.78 × 10^−2^

**Table 3 animals-09-00137-t003:** KEGG pathway enrichment of the DEGs in the Large White pigs from *GSE99092*.

Pathway ID	Name	Gene count	*p*-Value
**Upregulated DEGs**			
ssc01200	Carbon metabolism	17	5.04 × 10^−7^
ssc01100	Metabolic pathways	68	3.05 × 10^−6^
ssc00071	Fatty acid degradation	8	1.72 × 10^−4^
ssc01212	Fatty acid metabolism	8	2.37 × 10^−4^
ssc00280	Valine, leucine, and isoleucine degradation	8	3.73 × 10^−4^
ssc00640	Propanoate metabolism	6	5.48 × 10^−4^
ssc01210	2-Oxocarboxylic acid metabolism	5	8.14 × 10^−4^
ssc00630	Glyoxylate and dicarboxylate metabolism	5	2.26 × 10^−3^
ssc00350	Tyrosine metabolism	5	6.61 × 10^−3^
ssc00620	Pyruvate metabolism	5	1.08 × 10^−2^
ssc00061	Fatty acid biosynthesis	3	1.32 × 10^−2^
ssc03320	PPAR signaling pathway	6	2.76 × 10^−2^
ssc00190	Oxidative phosphorylation	9	3.24 × 10^−2^
ssc04920	Adipocytokine signaling pathway	6	3.37 × 10^−2^
ssc00360	Phenylalanine metabolism	3	3.52 × 10^−2^
ssc01230	Biosynthesis of amino acids	6	3.82 × 10^−2^
ssc04152	AMPK signaling pathway	9	3.83 × 10^−2^
ssc04975	Fat digestion and absorption	3	4.38 × 10^−2^
ssc00380	Tryptophan metabolism	4	4.55 × 10^−2^
**Downregulated DEGs**			
ssc04666	Fc gamma R-mediated phagocytosis	11	1.61 × 10^−3^
ssc04130	SNARE interactions in vesicular transport	7	2.63 × 10^−3^
ssc04512	ECM-receptor interaction	9	4.05 × 10^−3^
ssc00100	Steroid biosynthesis	5	4.94 × 10^−3^
ssc04974	Protein digestion and absorption	9	7.22 × 10^−3^
ssc04810	Regulation of actin cytoskeleton	15	2.87 × 10^−2^
ssc00310	Lysine degradation	6	3.41 × 10^−2^
ssc00260	Glycine, serine, and threonine metabolism	5	4.09 × 10^−2^
ssc00410	beta-Alanine metabolism	4	4.29 × 10^−2^
ssc04330	Notch signaling pathway	5	5.34 × 10^−2^
ssc00230	Purine metabolism	12	4.38 × 10^−2^
ssc00280	Valine, leucine, and isoleucine degradation	5	4.77 × 10^−2^
ssc00330	Arginine and proline metabolism	5	4.77 × 10^−2^

**Table 4 animals-09-00137-t004:** qRT-PCR array results for the AMPK signaling pathway (Large White-Min).

Gene Symbol	Fold Change	*p*-Value	Regulation
*CD36*	3.189283629	7.67 × 10^−3^	Up
*PGC-1A*	11.14850152	4.46 × 10^−2^	Up
*AKT2*	3.156554305	3.62 × 10^−3^	Up
*AKT1*	7.868832108	3.43 × 10^−5^	Up
*CPT1B*	8.890764573	3.93 × 10^−2^	Up
*ADIPOR2*	8.509277729	7.67 × 10^−3^	Up
*PPP2R1A*	6.599066891	1.04 × 10^−2^	Up
*CPT1A*	6.846403313	8.26 × 10^−4^	Up
*FOXO1*	5.397575689	3.18 × 10^−2^	Up
*LIPE*	6.924041605	1.87 × 10^−2^	Up
*LKB1*	4.901757799	7.41 × 10^−3^	Up
*FOXO3*	3.639771136	1.49 × 10^−4^	Up
*MTOR*	4.068698015	2.35 × 10^−2^	Up
*ADIPOR1*	2.245141118	3.62 × 10^−3^	Up
*RPS6KB2*	3.361252687	4.11 × 10^−2^	Up
*AKT3*	5.250417157	2.42 × 10^−2^	Up
*AMPK*	6.699341927	4.11 × 10^−2^	Up
*PDPK1*	1.981200539	7.58 × 10^−3^	Up
*TSC2*	2.740754696	7.67 × 10^−3^	Up
*SLC2A4*	9.440463793	1.01 × 10^−2^	Up
*CAMKK2*	5.359797456	2.16 × 10^−2^	Up
*TSC1*	8.141086731	1.08 × 10^−2^	Up
*PFKM*	−5.001312186	5.39 × 10^−3^	Down
*FBP2*	−5.808487321	1.08 × 10^−2^	Down
*ACC1*	−2.601477756	1.86 × 10^−2^	Down
*RAB2A*	−2.652531932	3.07 × 10^−3^	Down
*PFKFB1*	−6.096200009	1.47 × 10^−2^	Down
*PPP2R2A*	−4.510811417	3.62 × 10^−3^	Down
*IRS1*	−8.500540176	1.87 × 10^−2^	Down
*CREB3*	−5.026439516	1.03 × 10^−2^	Down
*PPP2R5A*	−5.729220232	1.53 × 10^−2^	Down
*PPP2R5B*	−3.011595315	7.67 × 10^−3^	Down
*PFKL*	−2.467192832	2.29 × 10^−2^	Down
*PIK3R2*	−3.841154966	2.35 × 10^−2^	Down
*PPP2R5E*	−1.634192954	3.77 × 10^−2^	Down
*CREB5*	−1.953383525	1.08 × 10^−2^	Down
*G6PC3*	−2.621247554	3.18 × 10^−2^	Down
*RAB11B*	−4.129252447	3.58 × 10^−2^	Down
*PIK3CB*	−2.137748295	4.37 × 10^−2^	Down
*PIK3CD*	−3.600875922	4.88 × 10^−2^	Down

**Table 5 animals-09-00137-t005:** Summary for the PPI network of 14 upregulated genes in the AMPK signaling pathway.

Gene Symbol	Degree
***SLC2A4***	9
***PGC-1α***	9
***LKB1***	6
***AMPK***	6
***LIPE***	6
***CD36***	5
***CAMKK2***	5
***FOXO3***	5
***FOXO1***	5
***CPT1B***	5
*CPT1A*	4
*ADIPOR2*	3
*ADIPOR1*	3
*PPP2R1A*	1

Top ten degree genes are shown in bold.

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
