# Peer review of "Data Mining and Validation of AMPK Pathway as a Novel Candidate Role Affecting Intramuscular Fat Content in Pigs"

_animals, 2019, doi:10.3390/ani9040137_

Round 1
Reviewer 1 Report
Authors need to describe the number of animals used for western boots and incorporate average data for the graphs along with standard deviations.
Incorporate protein data in results and discussion.
Author Response
Reviewer 1
Comments and Suggestions for Authors
Authors need to describe the number of animals used for western boots and incorporate average data for the graphs along with standard deviations.
Incorporate protein data in results and discussion.
Response to Reviewer 1 Comments
Dear Reviewer 1
Thank you for the helpful comments, which allowed us to improve our manuscript. We made the revisions accordingly. The main corrections and the responses to your comments are as follows:
Point 1: Authors need to describe the number of animals used for western boots and incorporate average data for the graphs along with standard deviations.
Incorporate protein data in results and discussion.
Response 1: We have added relevant information in Section 2.7 (L119) and Figure 6 (L207-L210) according to your suggestions. We have also appended protein data in the Results section (L189-L193) and Discussion section(L282-L288). We sincerely hope that you will appreciate and support our revision.
Figure 6. Relative expression levels of AMPK, LKB1, CaMKK2, CPT1A and ACC1 proteins in the AMPK signaling pathway in two pig breeds. All data are shown as the means ± SEM (n = 3), *p < 0.05, **p < 0.01.

Reviewer 2 Report
The manuscript was improved. I have no more comment.
Author Response
Reviewer 2
Comments and Suggestions for Authors
The manuscript was improved. I have no more comment.
Response to Reviewer 2 Comments
Dear Reviewer 2
We sincerely thank you for your appreciation and support for our revision. Your nice comments have helped us to improve our manuscript.

This manuscript is a resubmission of an earlier submission. The following is a list of the peer review reports and author responses from that submission.
Round 1
Reviewer 1 Report
Althouugh of interest, data is suported only at mRNA level by qPCR assays. Validation for protein expression levels is necessary and a major revision for this manuscript to be considered.
The authors need to specifically provide images of expression (western blot or immuostaining) for proteins deregulated compared to control for all their assays.Author Response
Reviewer 1
Comments and Suggestions for Authors
Although of interest, data is supported only at mRNA level by qPCR assays. Validation for protein expression levels is necessary and a major revision for this manuscript to be considered.
The authors need to specifically provide images of expression (western blot or immuostaining) for proteins deregulated compared to control for all their assays.
Response to Reviewer 1 Comments
Dear Reviewer 1
Thank you for your positive comments and valuable suggestions to improve the quality of our manuscript. The main corrections and the responses to your comments are as follows:
Point 1: The authors need to specifically provide images of expression (western blot or immuostaining) for proteins deregulated compared to control for all their assays.
Response 1: We agree with your nice comments and suggestions. We also realize that lack of protein experiments is a shortcoming for our study. However, due to the limited time for major revisions(10 days) and the initialization of the Spring festival holiday in China, we are unable to buy antibodies of all the hub proteins from the companies. As reported in previous studies, AMPK is a key regulator of the AMPK signaling pathway, LKB1 is the major ‘upstream’ activator of the energy sensor AMPK, and CPT1A is necessary for fatty acid oxidation and IMF deposition, and taking our results into account, we knew that the LKB1-AMPK-CPT1A axis play crucial role for fatty acid oxidation and IMF deposition in the LD muscle of Large White pigs. Therefore, we only added the western blot results of AMPK, LKB1 and CPT1A to further confirm our conclusions from detection of mRNA level. As shown in Figure 6, these three proteins showed higher expression in the LD muscle of Large White pigs than that in Min pigs. In addition, the western bolt analysis for other key proteins in the AMPK signaling pathway will be completed in our future studies. We sincerely hope that the current results of western blot will be appreciated and supported by you.
Figure 6. Relative expression levels of AMPK, LKB1 and CPT1A proteins in the AMPK signaling pathway in two pig breeds. All data are shown as the means ± SEM (n = 3), ***p < 0.001.

Reviewer 2 Report
Intramuscular fat content (IMF) is important to determine the meat quality. The authors investigated the signaling pathway that is associated with IMF content in muscle of white longissimus dorsi (LD) and the expression levels of the genes involved in the signaling pathways. They found that AMPK signaling may be associated with IMF in muscle of LD. Quantitative PCR analysis demonstrated that the expression level of some genes was changed between the muscles of LD of min and large white pigs. The results are novel, and may be useful for the recognition of meat quality of pigs. In addition, further analysis of molecular mechanism is interesting. I have some comments that should be addressed.
1. IMF and LD in Abstract should be written as the unabbreviated words when they are first appeared.
2. In Fig. 1D, size of inserted characters are too small. Please use bigger ones.
3. In the title on vertical axis, gene level should be expression level.
Author Response
Reviewer 2
Comments and Suggestions for Authors
Intramuscular fat content (IMF) is important to determine the meat quality. The authors investigated the signaling pathway that is associated with IMF content in muscle of white longissimus dorsi (LD) and the expression levels of the genes involved in the signaling pathways. They found that AMPK signaling may be associated with IMF in muscle of LD. Quantitative PCR analysis demonstrated that the expression level of some genes was changed between the muscles of LD of min and large white pigs. The results are novel, and may be useful for the recognition of meat quality of pigs. In addition, further analysis of molecular mechanism is interesting. I have some comments that should be addressed.
1. IMF and LD in Abstract should be written as the unabbreviated words when they are first appeared.
2. In Fig. 1D, size of inserted characters are too small. Please use bigger ones.
3. In the title on vertical axis, gene level should be expression level.
Response to Reviewer 2 Comments
Dear Reviewer 2
Thank you for the helpful comments, which allowed us to improve our manuscript. We made the revisions accordingly. The main corrections and the responses to your comments are as follows:
Point 1: IMF and LD in Abstract should be written as the unabbreviated words when they are first appeared.
Response 1: We have changed “IMF” to “Intramuscular fat”, and “LD” to “longissimus dorsi” in the Abstract section according to your suggestions(L13-L15).
Point 2: In Fig. 1D, size of inserted characters are too small. Please use bigger ones.
Response 2: Thanks for your suggestions. We have replaced the inserted characters with bigger ones in Figure 1D.
Figure 1. The Data mining results for the three GEO datasets. Scatter plot of DEGs in (A) GES24192, (B) GSE75045 and (C) GSE99092. (D) Venn diagrams of the KEGG pathways for the upregulated genes in the three GEO datasets. The six shared pathways (AMPK signaling pathway, the peroxisome proliferator-activated receptor (PPAR) signaling pathway, fat digestion and absorption, fatty acid metabolism, metabolic pathways and biosynthesis of amino acids) are listed in the figure. (E) Venn diagrams of the KEGG pathways for the downregulated genes in the three GEO datasets.
Point 3: In the title on vertical axis, gene level should be expression level.
Response 3: Thank you for your reminding. We have changed the “gene level” to “expression level” in the title of Figure 5.
Figure 5. Relative expression levels of ten key hub genes in the AMPK signaling pathway in two pig breeds. All data are shown as the means ± SEM (n = 3), ***p< 0.001.
